# Manifold Kernel Rank Reduced Regression

## Abstract

The Kernel Rank Reduced Regression (KRRR) technique works well on highly dependent dataset with a latent variable structure. When we extended the KRRR to the Reproducing Kernel Hilbert Space (RKHS), the powerful kernel presentation and reproducing ability can enhance the regression ability. But previous research always work on Euclidean space with vector data presentation, which omit the intrinsic geometric shape of the data distribution. If the whole dataset can be thought as a manifold, the regression result will only rely on the intrinsic data distribution instead of the extrinsic frame. So we present the manifold kernel rank reduced regression model (MKRRR). We fist give the definition of the MKRRR model. Then with leveraging Kendall shape space for representing sample manifold data, we derive the closed-form solution of the regression model and prediction result. Moreover, we discuss the convergent and robust ability of the model, with presenting the robustness proof. At last, the we present a skull repair application by the MKRRR model for 3D mandibular reconstruction. The experiment result validate effective of our model even on the data with high-level noise.

## 1 Introduction

Regression on the manifold is quite a difficult research for its irregular data presentation. Kernel Rank Reduced Regressing(KRRR) functions from Euclidean training data $\{(x_i, y_i)\}_{i=1}^{N}$ is well studied (Mukherjee & Zhu, 2011). Euclidean statistics do not describe the intrinsic structure of manifold-valued data well, and there may be corresponding errors in the results predicted by the model. However, the real shape is located on the low-dimensional submanifold, it is necessary to map the data to the Kendall shape space to get the intrinsic characteristics of the shape instead of in the 3D Euclidean space

We work on the manifold with the KRRR with different distance on manifold. We apply the manifold kernel regression method to analyze the internal relations of the data. In human skull reconstruction experiment, we compare the reconstruction effects with and without manifold measures. It is shown that considering manifold structure is beneficial to complex geometric objects. Finally, we map the point cloud data in the manifold space back to the original space. Figure1 describes the spatial transformation of the manifold kernel reduced-rank regression method in the operation process with an example of mandibular predictive reconstruction. **The contributions can be summarized as:**

(1) We present the present the manifold kernel rank reduced regression (MKRRR) model (Sec 3). The model obtains the intrinsic geometric distribution of high-dimensional nonlinear sample data. We construct the sample manifold data representation through Kendall shape space with compact data description by kernel regression. This algorithm is high accuracy, isometric invariance, and robustness to similarity transformations.

(2) We provide an close-form solution of the MKRRR(Sec 4.2). The close-form solution can control the precious of the regression result o the manifold. The algorithm is efficiency and smaller memory usage. It can work well on the complex and high-dimension model.

(3) We present a robustness proof (Appends A) of MKRRR. We get the condition that the model converges when the training data is disturbed. In the process of amplification and minification, we never damage the coefficient of the higher-order term. Therefore, this condition is a sufficient and necessary condition for model stability, not only a sufficient condition.

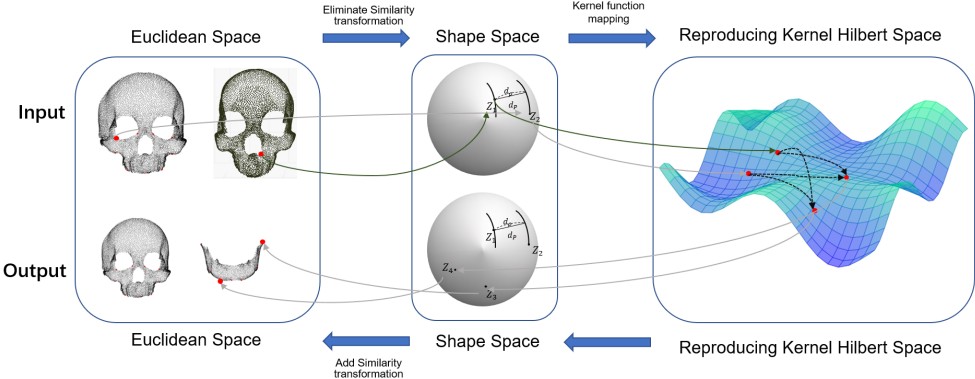

Figure 1: Schematic diagram of spatial transformation in the algorithm process

We give a application example of MKRRR model to reconstruct missing mandible bone in the 3D skull model which prove the effective mine ability of our model of the intrinsic correlation between complex typologies.

## 2 RELATED WORK

Rank Ridge Regression can help to produce a low-rank estimator of the regression coefficient matrix. This is very useful when the responses are highly dependent or there are reasons to believe a latent variable structure among the predictors. Ashin Mukherjee and Ji Zhu (Mukherjee & Zhu, 2011)extend the reduced rank idea to the RKHS set-up and give some intuition for the meaning of a rank constraint in a functional space. They get the solution to the Reduced Rank Ridge Regression problem which is as a projection of the Ridge Regression estimator to a constrained space. Ashin Mukherjee (Mukherjee, 2013) proposes a combination of the ridge penalty and rank constraint on the coefficient matrix. The ridge penalty helps to ensure that the estimate of the coefficient matrix is well-behaved even in the presence of multicollinearity, whereas the rank constraint encourages dimension reduction. Chen et al. (Chen et al., 2013) propose an adaptive nuclear norm penalization approach for low-rank matrix approximation, and use it to develop a new reduced rank estimation method for high-dimensional multivariate regression. Wu Qiong et al. Wu et al. (2020) propose an algorithm based on nuclear norm relaxation. A few numerical examples are presented to show the smaller mean squared prediction error compared with the elementwise univariate kernel ridge regression. These methods improve reduced rank ridge regression in kernel extension, rank constraint, and penalty terms.

Manifold-valued data, however, cannot be considered as a sample point in Euclidean sample space. So regression methods that originally worked for Euclidean data do not work for these data. Mapping such arbitrary output to a target manifold such as the 3D rotation space is not trivial, since $SO(3)$ is not homeomorphic to a Euclidean space (Brégier, 2021). Discretizing the target space is a common method, and rotationNet (Kanezaki et al., 2018) and SSD-6D (Kehl et al., 2017) have been successful in object attitude estimation. But their regression results are not accurate enough, as the number of classes required is typically of the order of $(1/\alpha^d)$ with respect to a typical discretization step $\alpha$ and the dimension d of the target manifold1 (d = 3 in this kind of method). Because their methods reformulate the regression into a classification problem in essence. Lin Lizhen et al. (Lin et al., 2017) propose an approach, that embeds the manifold where the responses lie onto a higher dimensional Euclidean space, obtains a local regression estimate in that space, and then projects this estimate back onto the image of the manifold. It can model data with manifold-valued responses and Euclidean predictors, but the regression framework is extrinsic.

## 3 MANIFOLD KERNEL REDUCED RANK REGRESSION

In this section, we will provide a detailed introduction to the problem formulation of manifold kernel reduced rank regression.

### 3.1 PROBLEM FORMULATION

In regression analysis, the input and output data can be seen as results of discretizing data from the original Euclidean space into a manifold representation. The input and output data are respectively represented as $x_i \in \mathbb{R}^p$ and $y_i \in \mathbb{R}^q$, where $p$ and $q$ represent the dimensions of the input and output data. The goal of regression analysis is to model the relationship between $x_i$ and $y_i$, with the purpose of predicting the output data $y_i$. However, since the data is high-dimensional and nonlinear, linear regression methods are not effective. To address this, the manifold-based kernel method is applied to project data from the manifold onto a reproducing kernel Hilbert space $\mathcal{H}$, where reduced rank regression analysis is performed. This approach improves the model's ability to handle nonlinear data and increases the accuracy and reliability of the regression analysis.

In the theory of reproducing kernel Hilbert spaces Suzuki (2022), it is guaranteed that for any positive definite kernel function $k$, there exists a feature mapping $\phi : E \to \mathcal{H}$, such that for any $x, y \in E$, $k(x, y) = \langle \phi(x), \phi(y) \rangle_{\mathcal{H}}$, where $\langle \cdot, \cdot \rangle_{\mathcal{H}}$ denotes the inner product operation of RKHS. (Manton et al., 2015) This mapping $\phi$ is known as the feature mapping of the kernel function $k$. With this in mind, the modeling problem described above can be represented in the following form:

$$
\begin{aligned}
y_{ij} &= f_j(x_i) + \varepsilon \\
y_i &= (y_{i1}, y_{i2}, \ldots, y_{iq})
\end{aligned}
\tag{1}
$$

where $f_j \in \mathcal{H}$ and $\varepsilon$ is the random error term. Then the objective of the model's regression analysis is to find a set of functions $(f_1, f_2, \ldots, f_q) \in \mathcal{H}$ that minimize the following loss function:

$$
L_\lambda(f_1, f_2, \ldots, f_q) = \sum_{i=1}^n \sum_{j=1}^q (f_j(x_i) - y_{ij})^2 + \sum_{j=1}^q \lambda \|f_j\|_{\mathcal{H}}^2
\tag{2}
$$

where $\|\cdot\|_{\mathcal{H}}$ denotes the norm of RKHS.

In order to utilize the correlation between the dependent variables yi, the rank constraint of the data in the reproducing kernel Hilbert space needs to be expressed in the form of rank constraint in an equivalent linear regression model, which satisfies the following:

$$
\dim(\text{span}\{f_1, f_2, \ldots, f_q\}) \leq r
\tag{3}
$$

where $1 \leq r \leq q$.

The solution to this problem is given in Section 4.

### 3.2 CONSTRUCTION OF MANIFOLD

The method proposed in this paper adopts a specific Riemannian manifold framework, which involves constructing a Kendall shape space Kendall (1984) to discretize the representation of the original data. As per Kendall's definition of shape space, shape space is a quotient space formed by configurations under the action of the similarity transformation group. In general, a configuration is a set of feature points for a specific object, and the configuration matrix can be represented as $X_{k \times m} = [l_1, l_2, \ldots, l_k]^T$, where $l_i \in \mathbb{R}^m$ represents a configuration composed of $k$ feature points with $m$ dimensions each.

Shape space, also known as Kendall shape space, is typically denoted as $\sum_m^k$, where $k$ and $m$ have the same meanings as in configurations. Based on the characteristics of Kendall shape space, its construction usually involves the following steps: 1) Translation transformation, 2) Scaling, and 3) Rotation elimination, which is included in the Appendix B.

### 3.3 ALGORITHMIC PROCESS

The problem model of MKRRR involves analyzing high-dimensional nonlinear data in Kendall shape space. The model aims to reduce the dimensionality of the data using the dimensionality reduction

characteristics of manifolds while incorporating relevant features of data in the Kendall shape space. The algorithmic process of MKRRR involves the following steps:

---

**Algorithm 1** Manifold Kernel Rank Reduced Regression

---

**Require:**
  1:   Training set data   $\mathbf{X}, \mathbf{Y}$
  2:   Number of similar samples   $k$
  3:   Predictors(The same kind of data as $\mathbf{X}$)   $\mathbf{x}'$
  4:   Parameters of the model: regularization parameter, rank constraint   $\lambda, r$;
**Ensure:**
  5:   responses   $\mathbf{y}'$;
  6: **for** $i = 1 : N$ **do**
  7:   $[x_i] \leftarrow shape\mathrm{Re}presentation(x_i)$
  8:   $[y_i] \leftarrow shape\mathrm{Re}presentation(y_i)$
  9: **end for**
 10: $[x'] \leftarrow shape\mathrm{Re}presentation(x')$
 11: $index \longleftarrow ([d_g([\mathbf{x}_1], [\mathbf{x}']), d_g([\mathbf{x}_2], [\mathbf{x}']), \ldots, d_g([\mathbf{x}_N], [\mathbf{x}'])])$         // $k$ samples that are most
     similar to $[\mathbf{x}']$ are selected as training data
 12: $\mathbf{X}, \mathbf{Y} \longleftarrow \mathbf{X}[index_{1:k}], \mathbf{Y}[index_{1:k}]$
 13: $[y'] \leftarrow kernelRRR(X, Y, [x'], \lambda, r)$         // Train the model and predict by MKRRR
 14: $\mathbf{y}' \leftarrow \beta [\hat{\mathbf{y}}'] R^{-1} + 1\gamma^T$
 15: **return $\mathbf{y}'$**

---

## 4 KERNEL AND ANALYTICAL SOLUTION OF MKRRR

One of the advantages of being able to compute positive definite kernels on manifolds is that it allows us to use algorithms developed for $\mathbb{R}^n$ space directly, while still taking into account the geometric shape of the manifold. In this section, we will discuss algorithms that use kernels induced by metrics on manifolds. MKRRR is a kernel-based reduced-rank regression statistical analysis method, where the choice of kernel function has a significant impact on the method. The main idea of kernel reduced-rank regression is to map the original low-dimensional data to a high-dimensional kernel space and then perform reduced-rank regression in the kernel space.

### 4.1 KERNEL AND CONSTRUCTION OF RKHS BASED ON MANIFOLD

The key challenge in extending kernel methods from Euclidean space to manifolds is to define appropriate positive definite kernel functions on the manifold. Currently, there is no direct way to extend Euclidean kernels (such as linear and polynomial kernels) to nonlinear manifolds. However, under certain conditions, the well-known Gaussian kernel in Euclidean space can be extended to manifolds in some way. In this paper, we define positive definite kernels based on Gaussian kernel functions on manifolds in the same way as in the literature (Turaga & Srivastava, 2015), enabling us to embed the given manifold into a high-dimensional RKHS with the corresponding metric. These positive definite kernels allow algorithms developed for original Euclidean space data to be extended to nonlinear manifold-valued data.

For various kernel-based algorithms, the Gaussian kernel has been shown to be very effective in Euclidean space. It maps the original data points to an infinite-dimensional Hilbert space, generating very rich data representations intuitively. In Euclidean space $\mathbb{R}^n$, the Gaussian kernel can be expressed as $k_G(x, y) := \exp(-\gamma||x - y||^2)$, using the Euclidean distance $||x - y||$ between two data points $x$ and $y$. Therefore, to define a kernel on a manifold, we need to use a more accurate distance metric on the manifold, the geodesic distance, to replace the Euclidean distance. However, not all geodesic distances on a manifold can produce positive definite kernels. For example, if the kernel function is defined as $\exp(-\gamma d_g^2(x, y))$ on the $n$-dimensional unit sphere embedded in $\mathbb{R}^{n+1}$, where $d_g$ is the geodesic distance or great circle distance commonly used on the manifold, the kernel function is negative definite. To distinguish it from the Gaussian kernel in Euclidean space, this paper refers to the Gaussian kernel defined by the distance metric on the manifold as a **Gaussian-type kernel**.

Table 1: Distance measurement and positive qualitative of Gaussian kernel in different Spaces

| Metric name | Formula | Geodesic distance | Positive define |
|---|---|---|---|
| **Euclidean Space** | | | |
| Euclidean distance | $d(x,y) = \sqrt{\sum_{i=0}^{n}(x_i - y_i)^2}$ | False | True |
| **Shape manifold** | | | |
| Full Procrustes | $d_{FP}(x,y) = \sqrt{1 - |\langle x,y \rangle|^2}$ | False | True |
| Partial Procrustes | $d_{PP}(x,y) = \sqrt{1 - |\langle x,y \rangle|}$ | False | False |
| Arc length | $d_\rho(x,y) = \arccos(|\langle x,y \rangle|)$ | True | False |

As introduced in kernel methods, when the kernel function is expressed in the form of the inner product of two data, the kernel function itself is positive definite. For the Gaussian-type kernel, let $(\mathcal{M}, d)$ represent the metric space composed of the manifold $\mathcal{M}$ and the metric $d$, and define the kernel $k : (\mathcal{M}, \mathcal{M}) \to \mathbb{R}$ on it as $k(x,y) := \exp(-\gamma d^2(x,y))$. Then, the kernel function $k$ is positive definite if and only if $d(x,y) = ||\phi(x) - \phi(y)||$ when $\gamma > 0$, where $\phi$ is a mapping from $\mathcal{M}$ to an inner product space $V$ and $\phi(x)$ is the result of mapping $x$ in $V$.

When considering the geodesic distance $d_g$, the Riemannian manifold $\mathcal{M}$ forms a metric space. Based on the aforementioned definition, it is natural to wonder under what conditions the geodesic distance on the manifold produces a positive definite Gaussian kernel. According to the theorem given by Jayasumana et al. (Turaga & Srivastava, 2015) for positive definite Gaussian kernels on manifolds, if the manifold $\mathcal{M}$ is isometric to some Euclidean space $\mathbb{R}^n$ in the Riemannian sense, then the Gaussian kernel defined by the geodesic distance on the manifold is positive definite. Although it is possible to find the Gaussian kernel induced by the geodesic distance on some Riemannian manifolds, it is theoretically impossible for other manifolds. In particular, if the manifold is compact, it is impossible to find an isometric mapping between the manifold and $\mathbb{R}^n$ because $\mathbb{R}^n$ is not compact. Therefore, it is not possible to obtain a positive definite Gaussian kernel from the geodesic distance on a compact manifold. In this case, the best hope is to find a different non-geodesic distance on the manifold that differs only slightly from the geodesic distance but still satisfies the conditions of the theorem.

Based on RKHS and the theorem on positive definite Gaussian kernels on manifolds, this paper uses geodesic distance and Procrustes analysis to define a positive definite kernel on shape manifolds, which are non-Euclidean manifolds representing the space of all possible shapes of a given object. The proposed method, called Geodesic Procrustes Analysis (GPA), is a kernel-based approach that embeds the shape manifold into an RKHS using the Gaussian-type kernel defined by the geodesic distance on the manifold.

Specifically, given a set of shapes on the manifold, the Procrustes analysis is used to bring them into a common coordinate system, which removes the differences in position, orientation, and scale. Then, the geodesic distance between the aligned shapes is used to define the Gaussian-type kernel, which is positive definite according to the theorem mentioned above. Finally, the embedded data points in the RKHS space can be used for various machine-learning tasks. In summary, the Geodesic Procrustes Analysis (GPA) method is a kernel-based approach that uses the geodesic distance and Procrustes analysis to define a positive definite kernel on shape manifolds. The analysis of different metrics in various spaces, whether they belong to geodesic distances, and the positive definiteness of their Gaussian kernels are summarized in Table 1.

## 4.2 ANALYTICAL SOLUTION OF MKRRR

According to the formulation of the problem in section 3.1, the objective of kernel reduced rank regression is to find the function $(f_1, f_2, \ldots, f_q) \in \mathcal{H}$ to minimize the Loss function equation 2, and at the same time, the constraint of equation 3.

According to the introduction of the Reproducing Kernel Hilbert Space (RKHS), we know that $\mathcal{H} := M \oplus M^\perp$, where

$$M := \text{span}(\{k(x_i, \cdot)\}_{i=1}^n)$$
$$M^\perp := \{f \in \mathcal{H} | \langle f, k(x_i, \cdot) \rangle_{\mathcal{H}} = 0, i = 1, 2, \ldots, n\}$$

(4)

Decompose each $f$ in RKHS, let $f = f^* + f^\perp$, $f^* \in M$, $f^\perp \in M^\perp$, then we have

$$f(x_i) = \langle f^* + f^\perp, k(x_i, \cdot) \rangle = f^*(x_i)$$
$$\|f\|_{\mathcal{H}}^2 = \|f^*\|_{\mathcal{H}}^2 + \|f^\perp\|_{\mathcal{H}}^2$$

(5)

Apparently, $L_\lambda(f_1^*, f_2^*, \ldots, f_q^*) \leq L_\lambda(f_1, f_2, \ldots, f_q)$, the objective of kernel reduced-rank regression can be transformed into optimizing $L_\lambda(f_1^*, f_2^*, \ldots, f_q^*)$. At the same time, since $f^*$ is the projection of $f$ in $M$ and $\dim(\text{span}\{f_1, f_2, \ldots, f_q\}) \leq r$, $\dim(\text{span}\{f_1^*, f_2^*, \ldots, f_q^*\}) \leq r$ still holds. Therefore, according to the representation theorem of RKHS, the optimization result of kernel reduced-rank regression 2 can be expressed as the following formula:

$$f_j(x) = \sum_{i=1}^n \alpha_{ij} k(x, x_i) \quad j = 1, 2, \ldots, q, \alpha_{ij} \in \mathbb{R}$$
$$[f_1, f_2, \ldots, f_q] = [k(\cdot, x_1), k(\cdot, x_2), \ldots, k(\cdot, x_n)] A_{n \times q}$$

(6)

Next, we need to find sufficient conditions, under which the rank constraint in equation 3 is equivalent to the rank constraint on the coefficient matrix $A = [\alpha_{ij}]_{n \times q}$, that is:

$$\dim(\text{span}\{f_1, f_2, \ldots, f_q\}) \leq r \Rightarrow \text{rank}(A) \leq r$$

(7)

After that, the parameters of the kernel reduced-rank regression model can be solved using the optimization method of linear reduced-rank regression models.

Since the reproducing kernel function $k(\cdot, \cdot)$ is strictly positive definite, and each sample point $x_i$ in the input data is distinct, when $r = q$, 7 clearly holds. When $r < q$, there must exist a linear combination $\sum_{i=1}^q a_i f_i \equiv 0$, which is equivalent to $\|\sum_{i=1}^q a_i f_i\|_{\mathcal{H}}^2 = 0$:

$$\left\| \sum_{i=1}^q a_i f_i \right\|_{\mathcal{H}}^2 = 0 \Leftrightarrow a_{q \times 1}^T A^T [(k(x_i, x_j))]_{i,j=1}^n A a_{q \times 1} = 0$$

(8)

Since $k(\cdot, \cdot)$ is strictly positive definite, 8 holds only when $Aa = 0_{q \times 1} \Leftrightarrow a \in Ker(A)$, where the coefficient matrix $A$ is obviously a mapping defined on the vector space $\mathbb{R}^q$. When $\dim(\text{Ker}(A)) > 0$, according to the rank-nullity theorem,

$$\text{rank}(A) + \text{nullity}(A) = q$$
$$\therefore \quad \text{rank}(A) + \dim(\text{Ker}(A)) = q$$
$$\therefore \quad \text{rank}(A) = q - \dim(\text{Ker}(A)) < q$$

(9)

From the above results, it can be seen that the rank constraint of the coefficient matrix $A$ is established. Therefore, by analogy with the solution method of linear reduced rank regression model, the solution of kernel reduced rank regression can be obtained. Recalling the solution of the reduced rank regression algorithm, for the input data $x \in \mathbb{R}^p$, given the rank constraint and regularization penalty $(\lambda, r)$, the prediction result of the model is:

$$\hat{Y}_x(\lambda, r) = x \left( X^T X + \lambda I \right)^{-1} X^T Y P_r$$

(10)

$P_r$ is the projection matrix formed by the $r$ principal eigenvectors of $P = Y^T X (X^T X + \lambda I)^{-1} X^T Y$. For Kernel Reduced Rank Regression, we use the Gram matrix $K$ instead of the inner product matrix $X^T X$ of the input data, we can get:

$$Y^T X (X^T X + \lambda I)^{-1} X^T Y \Rightarrow Y^T K (K + \lambda I)^{-1} Y$$
$$x \left( X^T X + \lambda I \right)^{-1} X^T Y \Rightarrow K(x) \left( K + \lambda I \right)^{-1} Y$$

(11)

Finally, the prediction form of Rernel Reduced Rank Regression is obtained as follows:

$$\hat{Y}_x(\lambda, r) = K(x)(K + \lambda I)^{-1} Y P_r^K$$

(12)

where $P_r^K$ is the projection matrix formed by the first $r$ principal eigenvectors of matrix $P$, and $K(x) = [k(x, x_1), k(x, x_2), \ldots, k(x, x_n)]$ is the similarity measure calculated by the kernel function $k$ between the predicted input $x$ and the training sample data.

## 5 EXPERIMENTS AND DATA ANALYSIS

The aforementioned method is also applied to the task of reconstructing 3D mandibular point clouds. Reconstruction of mandibular defects poses a challenging problem in digitally-assisted fine reconstruction due to the complex manifold structure of the human skull. We conducted experiments on optimizing the parameters of the MKRRR model and compared the reconstruction effect achieved by selecting training samples based on Kendall manifold with that obtained by randomly selecting samples without considering manifold metric, in order to validate the meaningfulness of our generalization of Kernel Rank Reduced Regression to manifolds. Additionally, we evaluated model robustness by predicting noise-disturbed samples.

### 5.1 DATASET

The data used in this experiment came from the Chinese craniofacial morphological information database of Beijing Normal University, which contains 215 sets of skull models, among which 126 sets of male skull models and 89 sets of female skull models. All sample volunteers were Chinese, aged 19-75. The processed three-dimensional skull model contains the same number of points, and each point corresponds to one by one. The upper skull point cloud contains 7,477 points and the lower jaw contains 2,340 points, The complete skull can be divided into the upper skull and the lower jaw to simulate the absence of the lower jaw, see figure2.

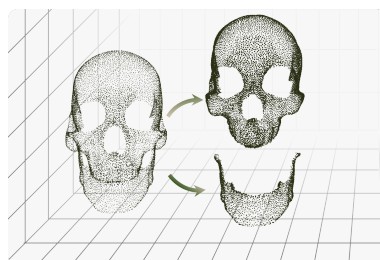

Figure 2: Original 3D skull model and its separation results

### 5.2 EVALUATION METRIC

From the practical point of view of 3D mandibular point cloud reconstruction, we pay more attention to the error of reconstruction results in Euclidean space. The kernel function we choose is full Procrustes distance Gaussian kernel $d_{FP}(x, y) = \sqrt{1 - |\langle x, y \rangle|^2}$. We use the relative errors mentioned in this paper Yan et al. (2022) to evaluate the reconstruction results, i.e. :

$$E_{eu} = \frac{\sum_{i=1}^{n} \left\| y_i - y_i' \right\|}{n \cdot \|b_{\max} - b_{\min}\|} \tag{13}$$

Where $b_{\max}$ and $b_{\min}$ are the largest and smallest vertices of the axially aligned bounding box, and $\|\cdot\|$ represents the Euclidean distance between two points.

In addition, in order to more accurately evaluate the reconstruction error after excluding the Similarity transformation, this paper calculates the distance between two shapes in the shape space to measure. Since the preshaped hypersphere is embedded in $^{(k-1)m}$, we consider the inferior arc length of the great circle between two points on the sphere as the shortest distance between the two shapes, called the Riemannian distance, denoted $d_p$, and use this geodesic distance to evaluate the similarity of the two shapes, i.e. :

$$E_{shape} = d(y, y') = \arccos(| < y, y' > |) \tag{14}$$

Our setting of $\gamma$ parameters will also follow the conclusions we obtained in Appendix A.

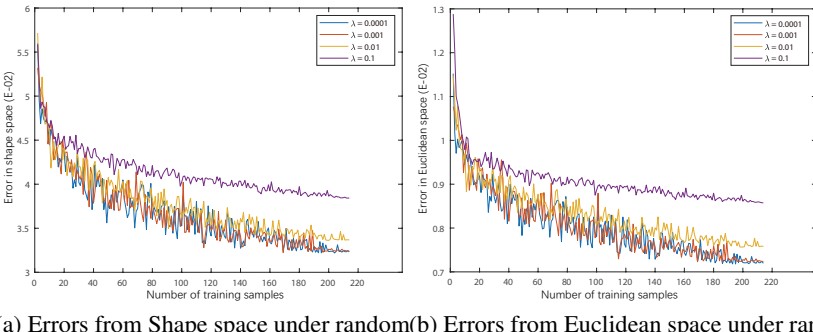

(a) Errors from Shape space under random training samples

(b) Errors from Euclidean space under random training samples

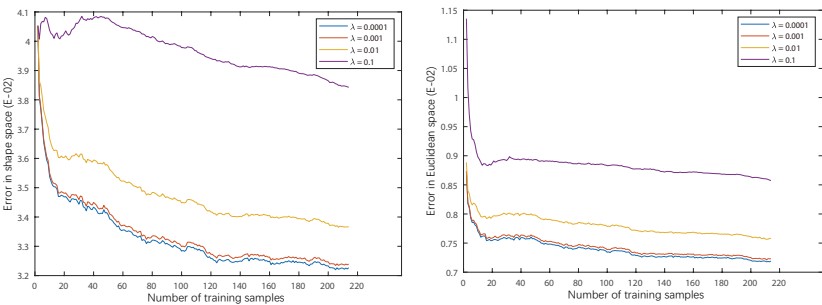

(c) Errors from Shape space under similar training samples

(d) Errors from Euclidean space under similar training samples

Figure 3: Regularization term $\lambda$ contrast, reconstruction error under different $\lambda$ values are compared.

### 5.3 RESULT ANALYSIS

As can be seen from figure 3, the reconstruction error is minimal when $\lambda$ is 0.0001.

In figure 4, when $1 < \alpha \leq 2$, the reconstruction error changes little with the change of rank constraint. In many experiments, it is found that when $\alpha = 1.2$, the reconstruction error reaches the minimum value, and the parameter is the optimal value. Finally, the optimal parameters of MKRRR we get are $\lambda = 0.00013$, $\alpha = 1.2$, $r = \min(p, q)/1.2$.

The results of the two experiments, one using random training samples and the other using similar training samples, demonstrate that models trained with similar training samples exhibit performance and lower reconstruction error compared to those trained with random training samples. Furthermore, as the number of training samples increases, the reconstruction results obtained from similar training samples become more stable. This highlights the importance of metrics on manifolds in our approach, particularly when dealing with objects possessing complex topological structures. Our method's advantage lies in its ability to effectively describe geometric object's topological relationships through manifold metrics.

As can be seen from figure 5, the MKRRR algorithm proposed in this paper shows better anti-noise performance, and the error range of its reconstruction results is relatively small. This shows that MKRRR algorithm has stronger robustness in processing noisy data and is expected to achieve good results in practical applications, Figure 7 shows the reconstruction effect.

## 6 CONCLUSION

In this paper, we illustrate the extension kernel regression to a Riemannian manifold framework. We provide the definition of MKRRR model. Subsequently, we compute the analytical solution for MKRRR on Kendall manifolds. Then we establish the robustness through proof of MKRRR model. Furthermore, we demonstrate the effectiveness of our method on a 3D mandib point cloud

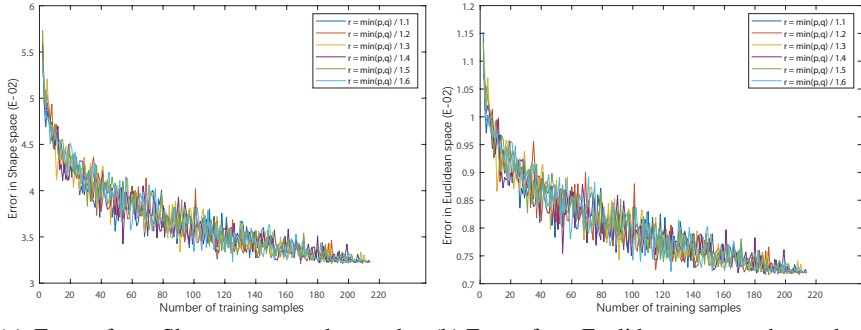

(a) Errors from Shape space under random training samples

(b) Errors from Euclidean space under random training samples

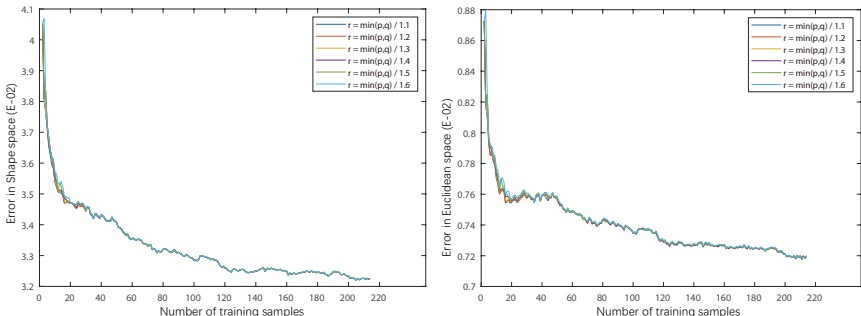

(c) Errors from Shape space under similar training samples

(d) Errors from Euclidean space under similar training samples

Figure 4: Regularization term $r$ contrast, reconstruction error under different $r$ values are compared.

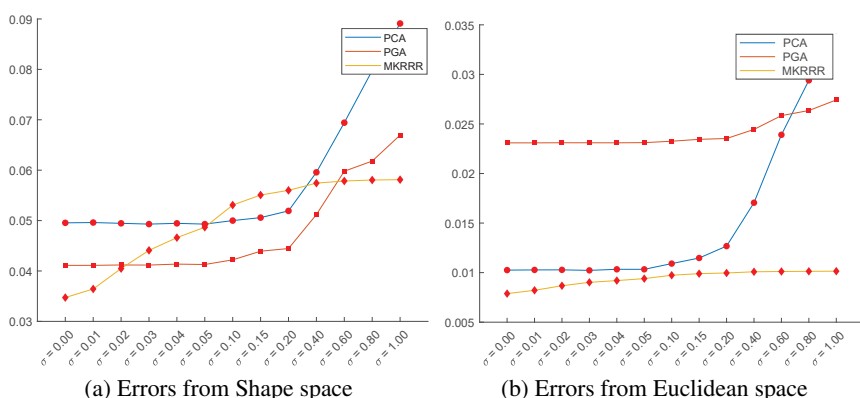

(a) Errors from Shape space

(b) Errors from Euclidean space

Figure 5: Comparison of reconstruction errors of different algorithms under different noise

dataset, which represents a complex topological structure, thereby showcasing its resilience. The experimental results suggest that Kernel Reduced Rank regression can be effectively employed on manifolds to yield meaningful estimates. By means of comparison, we highlight both the superiority of manifold metric and the necessity to extend the RKKK method to encompass manifolds. Based on the theory of regenerated kernel Hilbert space and the theorem of positive definite Gaussian kernel on manifolds, other distance measures can be explored to induce Gaussian kernel function and other manifold structure can be further discussed, which also leaves room for expansion of our method.

ETHICS STATEMENT

This section elucidates the concept of informed consent: The cranial data utilized in our experiment were acquired from human participants. All sample volunteers were Chinese individuals, ranging in age from 19 to 75 years old. Participants are provided with comprehensive information regarding the data collection process through written communication and are duly appr of potential risks and benefits associated with such collection We shall ensure utmost confidentiality of all personal information pertaining to participants, while also acknowledging their right to withdraw at any given point.

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

## A  ROBUSTNESS PROOF

Considering that 3D scanners have machine errors and often contain noise in the input data, we will prove the robustness of the model to noise in the following section. Because skull point cloud extraction needs to go through multiple data processing, its real noise source is complex, and Gaussian noise is the best simulation method for real noise. According to the central limit theorem, the superposition of a large number of independent random variable distributions tends to be a normal distribution.

The noise $\varepsilon = (\varepsilon(x))_{x \in X}$ are independent and identically distributed (i.i.d.) Gaussian random variables with mean 0 and standard deviation $\sigma > 0$. The expectations for the difference between noise-disturbed predictions and undisturbed predictions are as follows:

$$
\begin{aligned}
&E(|\widehat{Y}_{x+\varepsilon}(\lambda, r) - \widehat{Y}_x(\lambda, r)|)\\
=&E(|K(x+\varepsilon)(K+\lambda I)^{-1}YP_r^K - K(x)(K+\lambda I)^{-1}YP_r^K|)\\
=&E([|k(x+\varepsilon, x_1) - k(x, x_1)|, |k(x+\varepsilon, x_2) - k(x, x_2)|, \ldots, |k(x+\varepsilon, x_n) - k(x, x_n)|]) \cdot (K+\lambda I)^{-1}YP_r^K\\
=&[E(|k(x+\varepsilon, x_1) - k(x, x_1)|), E(|k(x+\varepsilon, x_2) - k(x, x_2)|), \ldots, E(|k(x+\varepsilon, x_n) - k(x, x_n)|)] \cdot (K+\lambda I)^{-1}YP_r^K
\end{aligned}
\tag{15}
$$

We choose Full Procrustes as the metric of the kernel function. Then we consider $|k(x+\varepsilon, x_i) - k(x, x_i)|$ where $i \in \{1, 2, \ldots, n\}$.

$$
\begin{aligned}
&|k(x+\varepsilon, x_i) - k(x, x_i)|\\
=&|\exp(-\gamma d^2(x+\varepsilon, x_i)) - \exp(-\gamma d^2(x, x_i))|\\
=&\exp(-\gamma d^2(x, x_i))|\exp(\gamma d^2(x, x_i) - \gamma d^2(x+\varepsilon, x_i)) - 1|
\end{aligned}
\tag{16}
$$

According to the definition of metric, in this kind of inner product space $|<x, y>|$ is bounded, so $d(x, y)$ is bounded. then we focus $|\exp[\gamma d^2(x, x_i) - \gamma d^2(x+\varepsilon, x_i)] - 1|$

$$
\begin{aligned}
&|\exp\{\gamma[d^2(x, x_i) - d^2(x+\varepsilon, x_i)]\} - 1|\\
=&|\exp\{\gamma[2|<\varepsilon, x_i>|\cdot|<x, x_i>| + |<\varepsilon, x_i>|^2]\} - 1|\\
\leq&|\exp\{\gamma[2|x_i|\cdot|<x, x_i>|\cdot|\varepsilon| + (|x_i|\varepsilon)^2]\} - 1|
\end{aligned}
\tag{17}
$$

So

$$
\begin{aligned}
&|\exp\{\gamma[d^2(x, x_i) - d^2(x+\varepsilon, x_i)]\} - 1|\\
\leq&2\int_0^{+\infty} |\exp[2\gamma|x_i|\cdot|<x, x_i>|\cdot\varepsilon + \gamma|x_i|^2\varepsilon^2] - 1|\cdot\exp(-\frac{\varepsilon^2}{2\sigma^2})\cdot d\varepsilon\\
\leq&2\int_0^{+\infty} \exp[2\gamma|x_i|\cdot|<x, x_i>|\cdot\varepsilon + \gamma|x_i|^2\varepsilon^2 - \frac{\varepsilon^2}{2\sigma^2}] + \exp(-\frac{\varepsilon^2}{2\sigma^2})\cdot d\varepsilon
\end{aligned}
\tag{18}
$$

This integral $\int_0^{+\infty} \exp(-\frac{\varepsilon^2}{2\sigma^2})d\varepsilon$ converges, and $\exp[2\gamma|x_i|\cdot|<x, x_i>|\cdot\varepsilon + \gamma|x_i|^2\varepsilon^2 - \frac{\varepsilon^2}{2\sigma^2}]$ is continuous over interval $[0, +\infty)$ and doesn't have improper point. Using one term Taylor expansion,

$$
\begin{aligned}
0\leq&\varepsilon^2\cdot\exp[2\gamma|x_i|\cdot|<x, x_i>|\cdot\varepsilon + \gamma|x_i|^2\varepsilon^2 - \frac{\varepsilon^2}{2\sigma^2}]\\
\leq&\exp[(2\gamma|x_i|\cdot|<x, x_i>| + 2)\cdot\varepsilon + (\gamma|x_i|^2 - \frac{1}{2\sigma^2})\varepsilon^2]
\end{aligned}
\tag{19}
$$

We should modify $\gamma < \frac{1}{2\sigma^2 |x_i|^2}$, so that

$$
\begin{aligned}
\lim_{x \to +\infty} \exp[(2\gamma|x_i| \cdot | < x, x_i > | + 2) \cdot \varepsilon + (\gamma|x_i|^2 - \frac{1}{2\sigma^2})\varepsilon^2] &= 0 \\
\lim_{x \to +\infty} \varepsilon^2 \cdot \exp[2\gamma|x_i| \cdot | < x, x_i > | \cdot \varepsilon + \gamma|x_i|^2\varepsilon^2 - \frac{\varepsilon^2}{2\sigma^2}] &= 0
\end{aligned}
\tag{20}
$$

There must exist a constant $\varepsilon_1 > 0$, when $\varepsilon > \varepsilon_1$,

$$
\begin{aligned}
|\varepsilon^2 \cdot \exp(2\gamma|x_i| \cdot | < x, x_i > | \cdot \varepsilon + \gamma|x_i|^2\varepsilon^2 - \frac{\varepsilon^2}{2\sigma^2}) - 0| &< 1 \\
0 \le \varepsilon^2 \cdot \exp(2\gamma|x_i| \cdot | < x, x_i > | \cdot \varepsilon + \gamma|x_i|^2\varepsilon^2 - \frac{\varepsilon^2}{2\sigma^2}) &< 1
\end{aligned}
\tag{21}
$$

And we know $\int_{\varepsilon_1}^{+\infty} \frac{1}{\varepsilon^2} d\varepsilon$ converge, so

$$
\int_{\varepsilon_1}^{+\infty} \exp[2\gamma|x_i| \cdot | < x, x_i > | \cdot \varepsilon + \gamma|x_i|^2\varepsilon^2 - \frac{\varepsilon^2}{2\sigma^2}]d\varepsilon
\tag{22}
$$

converge.

And $\exp[2\gamma|x_i| \cdot | < x, x_i > | \cdot \varepsilon + \gamma|x_i|^2\varepsilon^2 - \frac{\varepsilon^2}{2\sigma^2}]$ is continuous over interval $[0, \varepsilon_1]$ and doesn't have improper point and bounded. So $E|\exp[\gamma d^2(x, x_i) - \gamma d^2(x + \varepsilon, x_i)] - 1|$ converge.

And we can modify $\gamma < \frac{1}{2\sigma^2 |x_t|^2}$, $|x_t| = \min\{|x_1|, |x_2|, \ldots, |x_n|\}$. Then

$$
[E(|k(x + \varepsilon, x_1) - k(x, x_1)|), E(|k(x + \varepsilon, x_2) - k(x, x_2)|), \ldots, E(|k(x + \varepsilon, x_n) - k(x, x_n)|)]
\tag{23}
$$

converge.

Note that $(K + \lambda I)^{-1} Y P_r^K$ is a constant matrix, so the expectation of error converges and is bounded.

## B  MODEL DETAIL

### B.0.1  TRANSLATION TRANSFORMATION

The purpose of translation transformation is to eliminate the influence of spatial position on shape, as a shape only contains geometric information of configurations, which is invariant under position, rotation, and isotropic scaling (Euclidean similarity transformation).

The translation is the easiest to remove from configurations, and there are two ways to eliminate translation information: one is to move the center of the configuration to the origin, i.e., centering operation; the other is to remove the effect of the original position of the configuration by left-multiplying the matrix with a Helmert submatrix. The difference between the two methods is that the resulting matrix shape is different after removing the coordinate information. The former results in a $k \times m$ matrix after centering, while the latter results in a $(k - 1) \times m$ matrix. In this paper, we use the centering operation. The centering of configurations is generally achieved by subtracting the mean of the feature points from each feature point, i.e., $\bar{X} = \left[\frac{1}{k}\sum_{i=1}^{n} l_i, ..., \frac{1}{k}\sum_{i=1}^{n} l_i\right]$, and applying the following formula:

$$
\begin{aligned}
X_C &= X - \bar{X} \\
&= (I_k - \frac{1}{k}1_k 1_k^T)X \\
&= CX
\end{aligned}
\tag{24}
$$

This formula centers the configuration by subtracting the mean feature point from each feature point and multiplying the resulting matrix with $C$, which is a centering matrix.

### B.0.2 SCALING

Scaling is a method to eliminate shape size differences in shape space, which can be simply explained as making shapes in shape space have the same size. Scaling information is removed by dividing the configuration by the size of its centroid so that the scaled centroid has a size of 1. In this paper, we use the size of the centroid of the configuration $S(X)$ as the scaling factor:

$$S(X) = \|CX\| = \sqrt{\sum_{i=1}^{k} \left\| X_i - \bar{X} \right\|^2}$$
$$= \sqrt{trace(X^T CX)} \tag{25}$$

Once we obtain the size of the centroid, we can scale the configuration:

$$Z = \frac{1}{S(X)} CX = \frac{CX}{\|CX\|} \tag{26}$$

The result obtained at this point is called the pre-shape, which eliminates all information about position and proportion in the configuration. The pre-shape space is a quotient space formed by configurations of $k$ non-coincident pointsets in $R^m$ under the action of translation and isotropic scaling, and is typically denoted as $S_m^k$. The pre-shape space $S_m^k$ is a $(k-1)m$-dimensional unit radius hypersphere, where the norm of all pre-shapes in the space is 1. Pre-shape implies that it is one step away from the shape, i.e., the rotation information of the pre-shape is removed.

### B.0.3 ROTATION ELIMINATION

Rotation is the most variable information in a configuration and is the most difficult to remove. A $k \times m$ configuration $X$ can be rotated around the origin by left-multiplying with an $m$-dimensional rotation matrix $R$, which satisfies $R^T R = RR^T = I_m$ and $det(R) = +1$. The set of all $m$-dimensional rotation matrices is called the special orthogonal group $SO(m)$. The shape can be represented by the set $[X]$ as follows:

$$[X] = \{ZR : R \in SO(m)\} \tag{27}$$

Here, $Z$ is the pre-shape of configuration $X$, and $R$ is the rotation matrix. $SO(m)$ is an $m$-dimensional special orthogonal group.

To align the rotation of the pre-shape $Z$, Procrustes analysis is used, which expressed as follows:
$$R^* = \underset{R \in SO(m)}{argmin} \; d_g^2(Z_1 R, Z_2) \tag{28}$$

where $d_g(\cdot, \cdot)$ is the geodesic distance, which is the shortest distance between two shapes in shape space. According to the definition in (Dryden & Mardia, 2016), this distance is isometric to the arc length between two points on a hypersphere embedded in Euclidean space.

The alignment operation first selects a reference point and then aligns all shape data to the reference point. There are several ways to select a reference point, such as aligning all pre-shape data to the first pre-shape data, or calculating the intrinsic mean of all pre-shape data and aligning all pre-shape data to the mean shape. In this paper, we first align all data to the selected first pre-shape data, and then calculate the intrinsic mean of all pre-shape data, which is the average shape of the sample, and align all shapes to the intrinsic mean. Therefore, we also need to calculate the intrinsic mean of all pre-shape data.

First, Procrustes analysis is used to align the rotation of the pre-shape. The best rotation matrix $R$ that aligns all pre-shapes $Z_i$ to the first pre-shape $Z_1$ is calculated using Eq. 28. The aligned pre-shapes are obtained as $Z_i^{rot} = Z_i R$, and all pre-shapes aligned in the first step are denoted as $Z_1^{rot}, Z_2^{rot}, \ldots, Z_N^{rot} \in S_3^k$. Next, the intrinsic mean of all pre-shape data is calculated.

The concept of intrinsic mean was first proposed by Fréchet (M, 1948). Given a set of data $\mathbf{x}_1, \mathbf{x}_2, \ldots, \mathbf{x}_N \in \mathcal{M}$, Fréchet defined the intrinsic mean of the data as the shape $\mathbf{x}$in that minimizes the sum of squared geodesic distances to all shapes, i.e.:

$$\mathbf{x}_{\text{in}} = \arg \min_{\mathbf{x} \in \mathcal{M}} \sum_{i=1}^{N} d_g^2(\mathbf{x}, \mathbf{x}_i) \tag{29}$$

Clearly, the computation of intrinsic mean is a non-convex optimization problem, i.e., finding the minimum value of the squared distance function $f(x) = \frac{1}{2N} \sum_{i=1}^{N} d_g^2(\mathbf{x}, \mathbf{x}_i)$, which requires iterative optimization algorithms. We use the gradient descent algorithm adopted in (X, 1999; H, 1977) to iteratively compute the minimum value. The gradient of the function $f(x)$ is:

$$\nabla f(x) = -\frac{1}{N} \sum_{i=1}^{N} log_x(x_i) \tag{30}$$

Here, $log_x(x_i)$ represents the logarithmic mapping, and its norm can be used to represent the geodesic distance on the Riemannian manifold. Next, we choose the step size $\tau$ of the gradient descent algorithm and update the estimate $\mathbf{x}_{\text{t}}$ of the intrinsic mean using the following formula:

$$\mathbf{x}\text{t} + 1 = \exp\mathbf{x}\text{t}(\frac{\tau}{N} \sum i = 1^N log\mathbf{x}\text{t}(\mathbf{x}_i)) \tag{31}$$

The results of iterative optimization depend on the choice of the initial value and step size. Typically, the initial value for iteration is selected from the sample data, and the choice of the step size depends onthe manifold $\mathcal{M}$. Since the data is in pre-shape space, which is a unit-radius hypersphere composed of pre-shape trajectories, according to (R & P, 2001), a step size $\tau = 1$ on spherical data is sufficient for gradient descent convergence.

Thus, for $Z_1^{rot}, Z_2^{rot}, \ldots, Z_N^{rot}$, their intrinsic mean is:

$$Z_{\text{in}} = \arg \min Z \in S_3^k \sum i = 1^N d_g^2(Z, Z_i^{rot}) \tag{32}$$

After each update iteration, the result is:

$$Z_{\text{t}+1} = \exp Z\text{t}(\frac{1}{N} \sum_{i=1}^{N} log Z\text{t}(Z_i^{rot})) \tag{33}$$

It is worth noting that Kendall (Kendall, 1984) and Bhattacharya (Bhattacharya & Bhattacharya, 2012) pointed out that for Kendall's shape space $\sum_m^k$, when $m = 2$, the space is a compact differential manifold. However, for shape spaces $\sum_m^k$ where $m > 2$, they do not have a manifold structure. For shape spaces with $m > 2$, they have singularities (where configurations lie in subspaces of dimension $m - 2$ or smaller), which are called strata of the space. The existence of singularities breaks the differential structure of the manifold, and such spaces are called stratified spaces. In this study, we adopt the viewpoint proposed by Dryden et al. (Dryden & Mardia, 2016) as a prerequisite. That is, if the configuration data is modeled according to a continuous probability distribution, then the Lebesgue measure of the set of singularities is zero.

In general, the data found in nature usually follows a Gaussian distribution. Therefore, the assumption that there are no singularities in the shape space where shape data resides is also valid, and this assumption is usually reasonable in practice. This paper always assumes that the shape space is far away from the degenerate shapes that lead to singularities, and restricts the shape space to the manifold part.

# C  EXPERIMENTAL DETAILS

## C.1  EXPERIMENTAL PROCESS

In the reconstruction experiment, the regularization parameter $\lambda$ and the rank constraint term $r$ are the two decisive parameters that affect the reconstruction results in the MKRRR algorithm. Therefore, cross experiments are needed to verify the values of these two parameters before the comparison test, and the influence of the regularization parameter $\lambda$ and the rank constraint term $r$ on the algorithm is determined by Leave-One-Out Cross Validation respectively. Specific experimental methods are as follows:

Firstly, the choice of the regularization parameter $\lambda$ is usually between 0 and 1, and the regularization term exists to solve the problem of multiple collinearity in the training data, which leads to overfitting of the model. However, due to the large differences among the samples in the skull data set we adopted and the small amount of data in the data set, the value of the regularization term should not be set too large. Therefore, in the experiment, we mainly test the value range of $\lambda$ from 0.0001 to 0.1, and the value is expanded ten times each time. For each $\lambda$ value, we cross-validate 20 samples by Leave-One-Out method, and increase the training sample data one by one in step size 1 until all samples except the test sample data are used as training data. For each training sample number, we calculate its corresponding reconstruction error respectively, and finally take the average of all reconstruction errors as the reconstruction error under this $\lambda$ value. Finally, we choose the $\lambda$ value with the smallest reconstruction error as the final regularization parameter.

Secondly, the experimental verification method for rank constraint $r$ is similar to that for the regularization parameter $\lambda$, and the value range of rank constraint $r$ is $0 < r < \min(p, q)$, where $p$ and $q$ are the characteristic dimensions of the predictor variable and the response variable respectively. Let $r = \min(p, q)/\alpha$, in the experiment we test the reconstruction error in $1 < \alpha \leq 1.6$, $\alpha$ growth step is 0.1. The subsequent robustness comparison experiments under noise interference were carried out under optimal parameters obtained by us.

Thirdly, in order to verify the robustness of the method proposed in this paper, and considering the machine error of 3D scanner, we added Gaussian noise of different degrees to the input data. The noise standard deviation $\sigma$ was set within the range of 0.00 to 0.05, and the standard deviation increased by 0.01 each time. The reconstruction error of the algorithm under this condition is compared with the Example-oriented full mandible reconstruction based on principal component analysis proposed by Yan et al. (Yan et al., 2022) and 3D mandible reconstruction method based on PGA. Our setting of $\gamma$ parameters will also follow the conclusions we obtained in Appendix A. Gaussian noise is the best simulation method for skull point cloud data because the real noise source is complex after multiple data processing (Duan et al., 2014). According to the central limit theorem, the superposition of a large number of independent random variable distributions tends to normal distribution. Therefore, using Gaussian noise can better simulate the unknown real noise.

## C.2  PARAMETER ANALYSIS

As can be seen from figure 6, as the rank constraint r increases, the error of both measures decreases. It is difficult to determine the optimal value of resolving rank constraints from the two graphs, so we chose a smaller rank constraint for the experiment, which is shown in figure 4.

## C.3  RECONSTRUCTION EFFECT

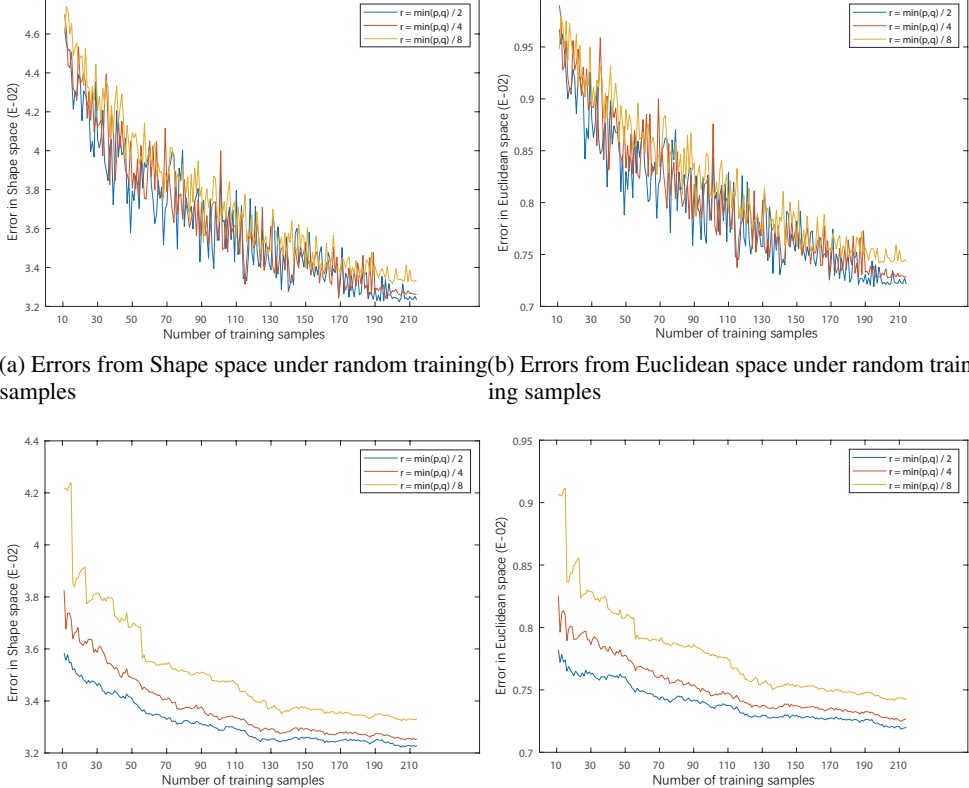

(a) Errors from Shape space under random training samples

(b) Errors from Euclidean space under random training samples

(c) Errors from Shape space under similar training samples

(d) Errors from Euclidean space under similar training samples

Figure 6: Regularization term $r$ contrast experiment, The reconstruction error results under different $r$ values are compared. Figure 6a shows the Shape space error comparison under random training samples, and Figure 6b shows the Euclidean space error comparison under random training samples. Figure 6c shows the Shape space error comparison under similar training samples, and Figure 6d shows the Euclidean space error comparison under similar training samples.

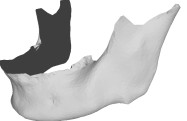 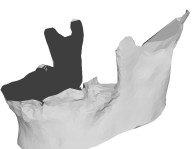

(a) An example of reconstructed mandibles      (b) Another example of reconstructed mandibles

Figure 7: Two example of reconstructed mandibles

