# OpenReview forum: "Manifold Kernel Rank Reduced Regression"
_ICLR.cc/2024/Conference — Submitted to ICLR 2024_

### Official Review · Reviewer_4QQN · 2023-10-23

**Soundness:** 2 fair
**Presentation:** 2 fair
**Contribution:** 1 poor
**Rating:** 3
**Confidence:** 3

**Summary:**

The paper introduces the Manifold Kernel Rank Reduced Regression (MKRRR) model as an extension of the Kernel Rank Reduced Regression (KRRR) technique. KRRR enhances regression capabilities for highly dependent datasets with latent variable structures. However, prior research focused on Euclidean space with vector data representation, neglecting the inherent geometric shape of data distribution. The MKRRR model addresses this issue and relies on the intrinsic data distribution, treating the dataset as a manifold.

The authors define the MKRRR model and derive a closed-form solution for regression and prediction. Finally, the authors present an application of the MKRRR model in 3D mandibular reconstruction for skull repair, demonstrating its effectiveness even on noisy data.

**Strengths:**

The authors apply MKRRR to 3D mandibular reconstruction which seems to be novel and original.

**Weaknesses:**

The authors present the KRRR as something novel, whereas it is already well-established in the community. For example, section 4.2 derives the closed form solution ok KRRR. This solution is already available in "Reduced Rank Ridge Regression and Its Kernel Extensions" A. Mukherjee et al. from 2011. The proof is very similar, and there is no novelty in the paper.

Furthermore, the MKRRR extension is simply using a Gaussian kernel and replacing the Euclidean distance by a Riemannian one. This should be stated in a clearer manner and more straightforwardly.

The application is not very convincing. The figures 3 and 4 are very noisy. The figure 4 is not readable and the figure 5 is barely presented. Hence, it is hard to draw conclusions from the experimental section. Furthermore, the hyper parameters seem to have been tuned on the test set.

**Questions:**

In the theoretical part:
1) What is new in your proof of the solution of KRRR compared to the one of "Reduced Rank Ridge Regression and Its Kernel Extensions" A. Mukherjee et al. from 2011?
2) Is MKRRR just replacing the Euclidean distance with a Riemannian one in the kernel?
3) Does a positive definite kernel with a Riemannian distance exist for your problem? The three proposed in table 1 are either not geodesic distances or not positive definite.

In the numerical experiments:

1) What do you mean by "each point corresponds to one by one"?
2) From figure 4, the rank value seems to have no effect on the performance. Can you comment on that?
3) Can you explain the comparison in figure 5? How do you compare PCA, PGA to MKRRR? The first two are only preprocessing steps while MKRRR does a regression.
4) What is the noise in figure 5? Do you add a Gaussian noise to data?

---

### Official Review · Reviewer_Vara · 2023-10-30

**Soundness:** 1 poor
**Presentation:** 1 poor
**Contribution:** 1 poor
**Rating:** 3
**Confidence:** 3

**Summary:**

The authors describe a method called manifold rank reduced regression and apply it on skull data.
The method combines elements from Riemannian geometry and linear algebra (reproducing kernel Hilbert space).
They study algorithmic aspects, provide a closed form solution and investigate robustness.

**Strengths:**

The authors seem to have a good knowledge of their domain.

**Weaknesses:**

The paper is not properly written.
The work lacks some general context and motivation and finally delve into an application for which the objectives are not clear either.
The authors should contextualize more their work and possibly provide more applications (and possibly mainstream applications that would appeal to a broader audience).

**Questions:**

What is the actual purpose of the experiment? Is it to recreate artificially a missing mandible?
If you focus on a specific application, how can the achieved accuracy be interpreted from the applicative side (archeology? are the archeologists satisfied with the numbers you achieve? or not?).
Can you clarify eq. 11? How do you justify going from left-hand side to right-hand side?

---

### Official Review · Reviewer_CQ8o · 2023-10-31

**Soundness:** 2 fair
**Presentation:** 3 good
**Contribution:** 3 good
**Rating:** 3
**Confidence:** 4

**Summary:**

This study expands upon the Kernel Rank Reduced Regression (KRRR) method, introducing it to the realm of the Kendall shape space manifold.

Nonetheless, the scope of the contribution appears somewhat constrained. The main result, the formula provided in equation (12) appears to have a generic applicability to various kernel regression approaches, rendering the results not particularly novel.

Additionally, one of the principal findings, pertaining to the robustness of our approach, lacks a rigorous mathematical definition, and I expect a theoretical analysis with more precise and well-defined explanations.

**Strengths:**

It aim to solve a problem with real applications.

**Weaknesses:**

1. the formula in (12) appears to be a standard solution, not novel.

2. no theoretical definition of the claimed "robustness" in the paper, and no rigorous proof as well.

**Questions:**

Could you please clarify the meaning of "shapeRepresentation" within the algorithm?

---

### Official Review · Reviewer_DLdA · 2023-11-01

**Soundness:** 2 fair
**Presentation:** 2 fair
**Contribution:** 2 fair
**Rating:** 3
**Confidence:** 3

**Summary:**

This work proposed manifold kernel rank reduced regression model (MKRRR) as the combination of both kernel rank reduced regression (KRRR) and geometric manifold structure. In particular this MKRRR utilize a specific type of Riemannian manifold framework, i.e., Kendall shape space.

The algorithm and the closed-form solution are present for MKRRR with the robustness proof in appendix. 3D point cloud reconstruction experimental results for Chinese skull models datasets are given with different settings and compared with few alternative methods.

**Strengths:**

The core idea of combining kernel rank reduced regression and manifold framework is convincing and make sense, and the introduced concept of Kendall shape space is interesting too.

Also good to see some related works been discussed in section 2. Closed-form solution is nice too and this additional robustness proof is helpful.

Reference 1: Q. Wu, F. M Wong, Y. Li, Z. Liu, and V. Kanade. Adaptive reduced rank regression. Advances in Neural Information Processing Systems, 33:4103–4114, 2020.

Reference 2: L. Yan, X. Wang, and Z, Wu. Example-oriented full mandible reconstruction based on principal component analysis. Multimedia Tools and Applications, 81(23):34009–34026, 2022

**Weaknesses:**

Equation 12 as the closed-form solution is nice and clean and should be helpful with more insight to present and how it is different from previous close-form solutions.

Good to see the experimental results in section 5 for real 3D data though seems with only 1 data set may not enough, and the comparison to alternative works limited to PCA and PGA seems less ideal, i.e., it should be good to include more.

It seems there are decent number of previous works in the area of kernel regression plus manifold learning, e.g., R3 is one of them and will be helpful if this work to try more comprehensive survey.

Reference 3: J. Nilsson, F. Sha, and M. I. Jordan. Regression on manifolds using kernel dimension reduction. International Conference on Machine Learning (ICML), pages 697–704, 2007

**Questions:**

Question 1: only Figure 5 with experimental results compared to alternative methods?

Question 2: section 2 "Ashin Mukherjee and Ji Zhu (Mukherjee & Zhu, 2011)extend" here need a space before "extend."

Question 3: section 3.1 "In order to utilize the correlation between the dependent variables yi" here yi seems need format change.

---

### Meta-Review · Area_Chair_E4Ni · 2023-12-01

**Metareview:**

This paper addresses the Kernel Rank Ridge Regression (KRRR) on the Riemannian manifold. The main criticism is on the claimed novelty, which has been an extension of existing literature. The key innovation and novelty claim need to be further clarified. The MKRRR algorithm requires additional clarification as well. In its current form, I would not recommend accepting the manuscript.

**Justification For Why Not Higher Score:**

The novelty of the paper is limited to be accepted in ICLR.

**Justification For Why Not Lower Score:**

N/A

---

### Decision · Program_Chairs · 2024-01-16

Reject